# Solving Single-objective tasks by preference multi-objective reinforcement learning

## Abstract

There ubiquitously exist many single-objective tasks in the real world that are inevitably related to some other objectives and influenced by them. We call such task as the objective-constrained task, which is inherently a multi-objective problem. Due to the conflict among different objectives, a trade-off is needed. A common compromise is to design a scalar reward function through clarifying the relationship among these objectives using the prior knowledge of experts. However, reward engineering is extremely cumbersome. This will result in behaviors that optimize our reward function without actually satisfying our preferences. In this paper, we explicitly cast the objective-constrained task as preference multi-objective reinforcement learning, with the overall goal of finding a Pareto optimal policy. Combined with Trajectory Preference Domination we propose, a weight vector that reflects the agent's preference for each objective can be learned. We analyzed the feasibility of our algorithm in theory, and further proved in experiments its better performance compared to those that design the reward function by experts.

## 1 Introduction

In recent years, Reinforcement Learning (RL) has achieved great success in many complex tasks, which commonly has a well-defined reward function (Mnih et al., 2015; 2016; Silver et al., 2016). However, there ubiquitously exists a type of task, which we call objective-constrained task, that is quite important but has not yet been well settled. As for the objective-constrained task, though only a single objective (denoted the primary objective in the following context) needs to be optimized, its difference from most RL scenarios lies in that there are some additional objectives in the environment. On one hand, in order to solve the primary objective better, it is necessary to optimize the additional objectives simultaneously. On the other hand, the additional objectives affect the primary objective more or less, whereas it is usually not clear how these additional objectives affect the primary objective. Take DOOM (Kempka et al., 2016) as an example, the primary objective of the agent is to kill as many enemies as possible. Meanwhile, there are additional objectives: picking up bullets and medicines, which may help kill more enemies in general, but their relationship with killing is still complex and ambiguous at every specific moment.

There have been two kinds of ways to solve the objective-constrained task. The first way focuses on the primary objective exclusively, hoping that the agent could learn more flexible policies. Take DOOM as an example, the environment gives back a reward whenever an enemy is killed by the agent (see Figure 1(a)). The deficiencies of this setting include: 1) Due to delayed reward (Arjona-Medina et al., 2018), it is difficult to find the relationship among the primary objective (killing enemies) and the additional objectives (picking up bullets and medicines). 2) Even if the agent finds that the additional objectives can help solve the primary objective better, the direction of policy updates may not be biased toward this behavior. The reason is that many timesteps are spent on these additional objectives without reward. Therefore, even if the cumulative return is greater, the cumulative discount return is even smaller. The phenomenon is called myopic policy (Bertsekas & Tsitsiklis, 1996), where a too low discount leads to highly sub-optimal policies. The second way is trying to clarify the relationship among these objectives using the prior knowledge of experts (see Figure 1(b)). One of the popular methods is to design a scalar reward that properly weighs the importance of each objective. However, this will often result in behaviors that optimize our reward function without actually satisfying our preferences (Russell, 2016; Everitt et al., 2017).

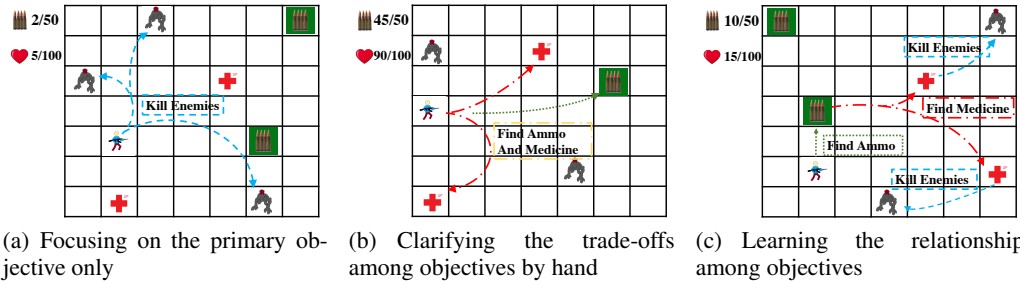

(a) Focusing on the primary objective only

(b) Clarifying the trade-offs among objectives by hand

(c) Learning the relationship among objectives

Figure 1: Comparison performance in DOOM between our algorithm and the other two mainstream works. (a) Due to the reward signal from killing only, it is difficult for the agent to find other objectives, even if they are useful for the task. (b) Trying to design the reward function using the prior knowledge of experts. This usually leads to the agent tending to get easy rewards rather than satisfying our preference. (c) Instead of applying excessive prior knowledge, a learnable weight vector is introduced to weigh the reward function of each objective and let the agent understand the purpose of the task.

Multi-objectivization (Knowles et al., 2001) is the process of transforming a single objective problem into a multi-objective problem. Recent research (Bleuler et al., 2001; Knowles et al., 2001) indicates that the methods from Pareto-based multi-objective optimization (MOO) may be helpful to solve single-objective optimization problems. In this paper, we reveal that transforming a single-objective reinforcement learning (SORL) to a multi-objective reinforcement learning (MORL) is beneficial for solving the primary problem. Rather than analyzing trade-offs of these objectives by hand, one of the keys of multi-objectivization is letting the agent learn the relationship among these objectives and understand the purpose of the task see Figure 1(c). It means that, instead of applying excessive prior knowledge, a learnable weight vector will be introduced to weigh the reward function of each objective. Different from obtaining a set of policies that approximate the Pareto front in MORL (Liu et al., 2017), we are merely interested in a Pareto optimal policy which fully reflects our preference information. Therefore, the known method (Brys et al., 2017) is not suitable for solving this task. The problem which only obtains some Pareto optimal policies in MORL is called **preference MORL**. The core idea underlying preference MORL is to modify the Pareto dominance relationship, so as to enhance the selection pressure of the algorithms and guide the algorithm to converge quickly to the preference region. Consequently, we propose a new measurement to estimate the agent's performance, which is called **Trajectory Preference Domination**. This can not only improve the efficiency of the algorithms to solve the optimization problems, but also reduce the computational complexity. In detail, instead of using a scalar return that combines all objectives, we adopt a new method of comparing agent's behavior where more than one measure is provided. Combined with the Trajectory Preference Domination, a weight vector that reflects the agent's preference for each objective can be learned. In this way, the learned reward function is better shaped. The weight vector assigns suitable preference to additional objectives and reduces the return of sub-optimal trajectories, so as to make the objective function of maximizing the cumulative undiscounted return consistent with the optimal policy. Through theoretical analysis, our algorithm effectively overcomes the problems of delayed reward and myopic policy in the objective-constrained task.

In summary, we solve the objective-constrained tasks from a completely new perspective of multi-objectivization. Our contributions are proposing a novel method to solve the problems and proving its feasibility in theory. Further, in order to identify the quality of our algorithm, we design a new benchmark problem: Efficient Delivery, which contains two additional objectives besides the primary objective. The experimental results show that our algorithm can learn the trade-off among these three objectives effectively and outperform those RL algorithms with the reward function designed by experts.

## 2 RELATED WORK

The constrained Markov Decision Process (Achiam et al., 2017; Horie et al., 2019) is a formulation for RL with constraints, where constraints on expectations of auxiliary costs must be satisfied. How-

ever, the relationship among these additional constraints and the primary objective is assumed to be known. The research in this field mainly focuses on acquiring policies that satisfy fixed constraints. Although there have been attempts to learn the reward function through inverse reinforcement learning (Abbeel & Ng, 2004; Ziebart et al., 2008), it is usually difficult to provide good demonstrations in complex tasks.

The idea of multi-objectivization has mainly been studied in the evolutionary computation works. There mainly exist two approaches for the multi-objectivization: either by decomposing the single objective (Handl et al., 2008; Jähne et al., 2009), or by adding extra objectives (Jensen, 2004; Brockhoff et al., 2007). Multi-objectivization through adding objectives typically involves the incorporation of some heuristic information or expert knowledge on the problem. The approach we propose in this paper falls in this category. Jensen (2004) is one of the pioneers to use what he calls helper objectives next to the primary one. He investigated the job-shop scheduling and traveling salesman problems and found that additional objectives based on time and distance-related intuitions respectively help solve these problems faster.

## 3 PRELIMINARY

RL (Sutton & Barto, 1998) is a paradigm that allows an agent to optimize its policy by interacting with a given environment. The agent is rewarded or punished for its behavior, and the aim is to maximize the accumulated reward over time. More formally, the environment is modeled as a Markov Decision Process (MDP) $< S, A, T, \gamma, R >$. $S = \{s_1, s_2, \dots\}$ is the state space of a finite set of states, and $A = \{a_1, a_2, \dots\}$ is the action space of a finite set of actions. When the environment is in state $s_t$, executing action $a_t$ makes the agent turn to state $s_{t+1}$ with probability $T(s_{t+1} \mid s_t, a_t)$, and a reward signal $r(s_t, a_t)$ is provided. Finally, $\gamma$, the discount factor, defines how important future rewards are. The goal of the agent is to find a policy $\pi$ that maximizes the expected cumulative discounted return $\mathcal{J}^\pi$.

$$\mathcal{J}^\pi = E\Big[ \sum_{t=0}^{\infty} \gamma^t R(s_t, a_t) \Big] \tag{1}$$

MORL (Liu et al., 2017) is a generalization of standard SORL, with the environment formulated as a MOMDP $< S, A, T, \gamma, \vec{R} >$. The difference between MORL and SORL lies in the reward function. In MORL, each objective has its own associated reward function, so the reward is not a scalar value but a vector:

$$\vec{R}(s, a) = \Big( R_1(s, a), \dots, R_m(s, a) \Big)$$

In MORL, policies are evaluated by their expected vector returns $\mathbb{J}^\pi$:

$$\mathbb{J}^\pi = \Big[ E\Big[ \sum_{t=0}^{\infty} \gamma^t R_1(s_t, a_t) \Big], \dots, E\Big[ \sum_{t=0}^{\infty} \gamma^t R_m(s_t, a_t) \Big] \Big] \tag{2}$$

Usually, these objectives often conflict with each other, so any policy can only maximize one of the objectives, or realize a trade-off among these conflicting objectives. In this case, it is difficult to order the candidate policies completely, and the concept of Pareto optimum is usually used: policy $\pi_1$ strictly Pareto dominates policy $\pi_2$, only if $\pi_1$ performs strictly better than $\pi_2$ at least on one objective and performs as well as $\pi_2$ on the other objectives. The set of non-dominated policies are referred to as the Pareto optimum set or Pareto front. The focus of MORL so far has been on developing new algorithms capable of finding a set of policies that approximate the Pareto front.

The most common approach (Moffaert et al., 2013; Vamplew et al., 2011) to deriving a policy is to optimize a linear scalarization reward function $\vec{w}(s, a) \cdot \vec{r}(s, a)$ based on a weight vector $\vec{w}$. The weight vector expresses which trade-off solutions the decision-makers prefer. However, it is often non-intuitive (Das & Dennis, 1997) for preference elicitation through setting the weight vector, and often requiring significant amounts of parameter tuning processes.

## 4    THEOREM OF PREFERENCE MORL

To perform multi-objectivization, we must add some helper-objectives to the primary single-objective problem. More precisely, the reward function is not a scalar value $r_p(s_t, a_t)$ but a vector $\vec{r_t} = [r_p(s_t, a_t), r_{h_1}(s_t, a_t), \ldots, r_{h_n}(s_t, a_t)]$. Generally speaking, $r_i = 1$ ($i = p, h_1, \ldots, h_n$) is provided when the agent accomplishes the $i$th objective, otherwise $r_i = 0$. Moreover, we should ensure that the optimal policy of the primary SORL is one of the policies in Pareto front of the corresponding MORL.

$$\forall \pi^\star, \exists \pi' \in \Pi^m, \pi' = \pi^\star,$$

where $\pi^\star$ is an optimal policy to the SORL problem, and $\Pi^m$ is the set of Pareto optimal policies.

Typically, the helper-objectives are in conflict with the primary objective, at least for some parts of the search space. Therefore, it is necessary to design a scalar reward that properly weights the importance of the primary objective and helper-objectives so that the intention of the task is reflected. In our work, a kind of measures is allowed to provide feedback on our agent's behavior and use this feedback to learn a weight vector in order to reflect the intention of the primary problem. Specifically, a new domination criterion is designed to compare the agent's behavior.

**Definition 1.** *(Trajectory Preference Domination): A full trajectory $\tau^1 = \left((s_0^1, a_0^1), \ldots, (s_{k-1}^1, a_{k-1}^1)\right)$ preference dominates another full trajectory $\tau^2 = \left((s_0^2, a_0^2), \ldots, (s_{t-1}^2, a_{t-1}^2)\right)$, denoted as $\tau^1 \succ_{pf} \tau^2$, if and only if one of the following two conditions is satisfied.*

(1) $\tau^1$ Pareto dominates $\tau^2$:

$$\forall i, \sum_k r_i(s_k^1, a_k^1) \geq \sum_t r_i(s_t^2, a_t^2) \quad \wedge \quad \exists i, \sum_k r_i(s_k^1, a_k^1) > \sum_t r_i(s_t^2, a_t^2).$$

(2) $\sum_k r_p(s_k^1, a_k^1) > \sum_t r_p(s_t^2, a_t^2)$.

In this way, Trajectory Preference Domination can be used to evaluate the agent's behavior quantitatively. Obviously, if a full trajectory $\tau^1$ is better than another full trajectory $\tau^2$ in the primary SORL problem, then using this preference domination criterion in the corresponding MORL problem can still ensure that $\tau^1$ is better than $\tau^2$. So optimizing the reward function produced by Trajectory Preference Domination can guide the agent to quickly converge to the preference multi-objective region, which is also preferred by the primary problem.

More precisely, the process of learning reward function can be modeled in the following way:

if

$$\left((s_0^1, a_0^1), \ldots, (s_{k-1}^1, a_{k-1}^1)\right) \succ_{pf} \left((s_0^2, a_0^2), \ldots, (s_{t-1}^2, a_{t-1}^2)\right),$$

then

$$r(s_0^1, a_0^1) + \cdots + r(s_{k-1}^1, a_{k-1}^1) \gg r(s_0^2, a_0^2) + \cdots + r(s_{t-1}^2, a_{t-1}^2),$$

where $r(s_t, a_t) = \sum_i w_i(s_t, a_t) \cdot r_i(s_t, a_t)$, and without losing its generality $w_i(s_t, a_t) \in [0, 1]$. The weight vector determines that which objective the agent should achieve at a specific state.

**Definition 2.** $<s, a>$ *is the preference state-action pair* $Pre(s, a)$ *if* $\exists i, r_i(s, a) = 1, w_i(s, a) > 0$. $<s, a>$ *is the non-preference state-action pair* $nPre(s, a)$ *if* $\forall i, if\ r_i(s, a) = 1, then\ w_i(s, a) = 0$. *And,* $<s, a>$ *is trivial state-action pair* $Tri(s, a)$ *if* $\forall i, r_i(s, a) = 0$.

Obviously, from the expression, we can see that the $Pre(s, a)$ is the state-action pair that should be contained in the optimal trajectories. Any trajectory containing $nPre(s, a)$ cannot complete the tasks. $Tri(s, a)$ is the state-action pair that does not achieve any objectives.

**Definition 3.** *The optimal trajectory cluster $T$ is a set of trajectories containing all preference state-action pairs but no non-preference state-action pairs.*

**Lemma 1.** *Starting from a state $s$, then the accumulated reward along $\tau$ is far greater than the accumulated reward along $\tau'$, where $\tau \in T, \tau' \notin T, s \in \tau$.*

Proof by contradiction:

$$\forall \tau \in T, \tau = \left( (s_0^1, a_0^1), (s_1^1, a_1^1), \ldots, (s_i^1, a_i^1), \ldots, (s_{k-1}^1, a_{k-1}^1) \right),$$

$$\exists \tau' \notin T, \tau' = \left( (s_0^1, a_0^1), (s_1^1, a_1^1), \ldots, (s_i^1, a_i^2), \ldots, (s_{t-1}^2, a_{t-1}^2) \right).$$

Suppose that Lemma 1 is false, then:

$$r(s_i^1, a_i^1) + \sum_{m=i+1}^{k-1} r(s_m^1, a_m^1) \le r(s_i^1, a_i^2) + \sum_{n=i+1}^{t-1} r(s_n^2, a_n^2),$$

$$\sum_{m=0}^{i-1} r(s_m^1, a_m^1) + r(s_i^1, a_i^1) + \sum_{m=i+1}^{k-1} r(s_m^1, a_m^1) \le \sum_{n=0}^{i-1} r(s_n^1, a_n^1) + r(s_i^1, a_i^2) + \sum_{n=i+1}^{t-1} r(s_n^2, a_n^2), \quad (3)$$

However, $\tau \succ_{pf} \tau'$, then:

$$\sum_{m=0}^{i-1} r(s_m^1, a_m^1) + r(s_i^1, a_i^1) + \sum_{m=i+1}^{k-1} r(s_m^1, a_m^1) \gg \sum_{n=0}^{i-1} r(s_n^1, a_n^1) + r(s_i^1, a_i^2) + \sum_{n=i+1}^{t-1} r(s_n^2, a_n^2). \quad (4)$$

Equation 3 is in conflict with equation 4, therefore, Lemma 1 is true.

**Theorem 1.** *Suppose that the reward function has been learned, then the policy that aims to maximize the accumulated discounted reward over time can also maximize the accumulated reward. And the quality of the policy is independent of discount factor $\gamma$.*

Proof:

Considering the computational form of the Q-function, we prove it from the terminate state.

$\forall \tau \in T, \tau' \notin T$, take the last preference state-action pair $< s_{t-1}, a_{t-1}^\tau >$ from $\tau$, and we know from the Lemma 1:

$$r(s_{t-1}, a_{t-1}^\tau) > r(s_{t-1}, a_{t-1}^{\tau'}) + r(s_t, a_t^{\tau'}) + \ldots$$

$$r(s_{t-1}, a_{t-1}^\tau) > r(s_{t-1}, a_{t-1}^{\tau'}) + \gamma \cdot r(s_t, a_t^{\tau'}) + \ldots, 0 < \gamma < 1$$

$$\therefore \max_i Q(s_{t-1}, a_{t-1}^i) = Q(s_{t-1}, a_{t-1}^\tau)$$

Take the next-to-last preference state-action pair$< s_{t-i}, a_{t-i}^\tau >, i > 1$ from $\tau$. According to the Lemma 1, starting from $s_{t-i}$, we know that the accumulated reward along $\tau$ is greater than the accumulated reward along $\tau'$.

It is possible that $< s_{t-i}, a_{t-i}^\tau > \in \tau'$, then:

$$r(s_{t-i}, a_{t-i}^\tau) \ge r(s_{t-i}, a_{t-i}^{\tau'}), \quad if \ a_{t-i}^\tau = a_{t-i}^{\tau'}$$

According to Definition 2, the reward value of those trivial state-action pairs and non-preference state-action pairs is 0. Add those trivial state-action pairs between $s_{t-i}$ and $s_{t-1}$, so as to complete the reward over trajectory.

If $s_{t-1} \in \tau'$, then:

$$r(s_{t-i}, a_{t-i}^\tau) + 0 + \cdots + 0 + r(s_{t-1}, a_{t-1}^\tau)$$
$$> r(s_{t-i}, a_{t-i}^{\tau'}) + 0 + \cdots + 0 + r(s_{t-1}, a_{t-1}^{\tau'}) + r(s_t, a_t^{\tau'}) + \ldots$$

When choosing a trajectory that the time-steps from $s_{t-i}$ to $s_{t-1}$ is shortest, then:

$$r(s_{t-i}, a_{t-i}^\tau) + \gamma \cdot 0 + \cdots + \gamma^{m-1} \cdot 0 + \gamma^m \cdot r(s_{t-1}, a_{t-1}^\tau)$$
$$> r(s_{t-i}, a_{t-i}^{\tau'}) + \gamma \cdot 0 + \cdots + \gamma^{m-1} \cdot 0 + \gamma^m \cdot r(s_{t-1}, a_{t-1}^{\tau'}) + \gamma^{m+1} \cdot r(s_t, a_t^{\tau'}) + \ldots$$

$$\therefore \max_j Q(s_{t-i}, a_{t-i}^j) = Q(s_{t-i}, a_{t-i}^\tau) \tag{5}$$

Similarly, if $s_{t-1} \notin \tau'$ or $< s_{t-i}, a_{t-i}^\tau > \notin \tau'$, obviously equation 5 holds.

For the same reason, we can prove that the Q-value of any preference state-action pairs is larger than the Q-value with other actions. And, the Q-value reaches the maximum when the corresponding actions make the shortest timesteps from any trivial state to the next preference state. Therefore, the policy that aims to maximize the accumulated discounted reward over time can also maximize the accumulated reward. Moreover, we do not make any assumptions about discount factor $\gamma$ in the proof, so the quality of the policy is independent of $\gamma$ in theory if exploration tends to infinite sufficiency. In addition, the weight vector assigns preference to helper-objectives that are typically helpful for exploring better solutions in the primary objective.

# 5 ALGORITHM OF PREFERENCE MORL

After converting the single-objective problem to a preference multi-objective problem, a policy $\pi : s \to a$ and a weight estimate $W : s \times a \to \vec{w}$ are parametrized by the neural network, respectively.

The two networks are updated iteratively by three processes:

1. The policy $\pi$ interacts with the environment with an exploration rate $\varepsilon$ to produce a set of full trajectories $\{\tau^1, \ldots, \tau^m\}$. Each trajectory $\tau^i$ needs to evaluate its cumulative undiscounted return for each objective and then puts the trajectory with its evaluation $e^i$ into a trajectory buffer $\mathcal{D}_{trj}$.

2. Randomly select pairs of trajectories $(\tau^i, \tau^j)$ from the buffer $\mathcal{D}_{trj}$, and compare their evaluation using Trajectory Preference Domination, then the parameters of the mapping $W$ are optimized via supervised learning to fit the comparisons.

3. The parameters of $\pi$ are updated by a traditional reinforcement learning algorithm, in order to maximize the discount sum of predicted rewards $r(s_t, a_t) = \sum_i w_i(s_t, a_t) \cdot r_i(s_t, a_t)$.

## 5.1 SETTING TRAJECTORY BUFFER

It is obvious that there must be lots of different trajectories for learning a true weight vector. An important problem is that what kind of buffer is more suitable for learning reward function. Making the replay buffer larger so that early trajectories obtained through random exploration are still present is impractical for two reasons; (1) Unless the buffer is infinite, older trajectories will eventually be erased before acquiring the true reward function. (2) Even if all past trajectories are still present, learning from an increasing number of trajectories remains hard, because the learning speed of each iteration will be slower and slower. Therefore, instead of retaining all past trajectories, the following two steps are taken to ensure effective learning of weight vector: 1) retaining top $K$ preference optimal trajectories in the current buffer; 2) replacing the remaining $M - K$ trajectories in the current buffer with new trajectories sampled by the current round. The purpose of the first step is to improve the utilization rate of preference trajectories and push reward learning towards the preferred direction. The second step can enable the reward function to be effectively learned in the case of the limited buffer. In addition, when there is a greater diversity of trajectories, it is more helpful for the reward learning and larger exploration rate can be realized.

## 5.2 OPTIMIZING THE WEIGHT VECTOR AND THE POLICY

We can interpret the weight estimation $W$ as a preference predictor if we view $w$ as a latent factor explaining which objective the agent should prefer at a specific time and state. Assume that the probability of preferring a trajectory $\tau^i$ depends exponentially on the value of the latent reward summed over the length of the trajectory:

$$P[\tau^1 \succ_{pf} \tau^2] = \frac{exp \sum \sum_i w_i(s_t^1, a_t^1) \cdot r_i(s_t^1, a_t^1)}{exp \sum \sum_i w_i(s_t^1, a_t^1) \cdot r_i(s_t^1, a_t^1) + exp \sum \sum_i w_i(s_t^2, a_t^2) \cdot r_i(s_t^2, a_t^2)}. \tag{6}$$

We optimize $w$ to minimize the cross-entropy loss:

$$loss(w) = - \sum_{\{(\tau^1, e^1), (\tau^2, e^2)\} \subset \mathcal{D}} \Big[ I(e^1 \succ_{pf} e^2) \cdot logP[\tau^1 \succ_{pf} \tau^2]$$

$$+ I(e^2 \succ_{pf} e^1) \cdot logP[\tau^2 \succ_{pf} \tau^1] \Big], \tag{7}$$

where $I(e^1 \succ_{pf} e^2) = \begin{cases} 1 & if \ e^1 \succ_{pf} e^2, \\ 0 & otherwise. \end{cases}$

Equation 6 and 7 are very similar to preference learning proposed by Christiano et al. (2017). The main difference lies in the comparison between trajectories, the latter used a large amount of human prior knowledge, which is usually related to tasks. Instead, Trajectory Preference Domination we proposed is agnostic to tasks. After learning weights $w$, the reward function $r(s_t, a_t) = \sum_i w_i(s_t, a_t) \cdot r_i(s_t, a_t)$ can be used to optimize the policy $\pi$. We can train this policy using any RL algorithm that is appropriate for the domain. In this paper, Deep Q-learning (DQN) (Mnih et al., 2013) is adopted.

# 6 EXPERIMENTS AND RESULTS

## 6.1 EFFICIENT DELIVERY

To identify the quality of our algorithms, we design a new benchmark problem (Figure 2): Efficient Delivery. In this game, the purpose of the agent is to control an unmanned aerial vehicle (UAV) to deliver as many packages as possible in the city (represented by 10*10 grid world). When the current delivery location is reached, the next delivery location appears at some location of the city. However, because of its limited battery capacity, the UAV cannot fly for too long. Therefore, a charging station is set in the upper left corner of the grid world. In this case, the agent needs to weigh the importance of delivery and charging at every moment so as to deliver as many packages as possible in a fixed period of time (100 timesteps). Moreover, in order to make the task more interesting and more challenging, an accelerator appears at the same time as the current delivery location appears, but their locations are different. Once the agent gets an accelerator, its flight speed will double in a short time (8 timesteps) and the accelerator disappears until the next delivery location appears. In detail, when the current delivery and accelerator locations are relatively close, even though it is a longer distance for the agent to obtain the accelerator firstly and then fly to the current delivery location than to reach the current delivery location directly, the continuous acceleration may make the agent reach the next delivery location faster. However, when the current accelerator location is far away from the delivery location, it is not worth the effort to fly through such a long distance to find the accelerator for short-term acceleration. In summary, in order to deliver as many packages as possible in a fixed period of time, it is necessary for the agent to weigh the importance of the three objectives at every moment: delivery, charging and acceleration.

## 6.2 RESULTS

We test the performance of our algorithm in two different settings. The first is that there are only two objectives: delivery and charging, whose importance needs to be clearly weighed. The second is that preferences for all three objectives need to be optimized. Except for the number of objectives, all hyper-parameters are the same in two settings. The configuration of these hyper-parameters is as follows: $\gamma = 0.9$, the size of trajectory buffer $M = 300$. The baseline is DQN optimized by the reward function which is provided only from the primary objective–delivery.

**Algorithm Effectiveness Testing.** To solve the delivery task, the agent must learn to weigh the importance of the three objectives at every moment. However, the existing methods require the task-specific prior knowledge such as a well-defined reward function (Mnih et al., 2015; 2016) or demonstration samples (Abbeel & Ng, 2004). To our knowledge, our work is the first one to solve such tasks through a learning method. As shown in Figure 3(a), our algorithm can make full use of the objectives of charging and acceleration to boost up the performance of the delivery task. The number of delivery is obviously the most in the situation with all three objectives, compared to the situation with two objectives or with the reward of delivery only. This result not only shows the

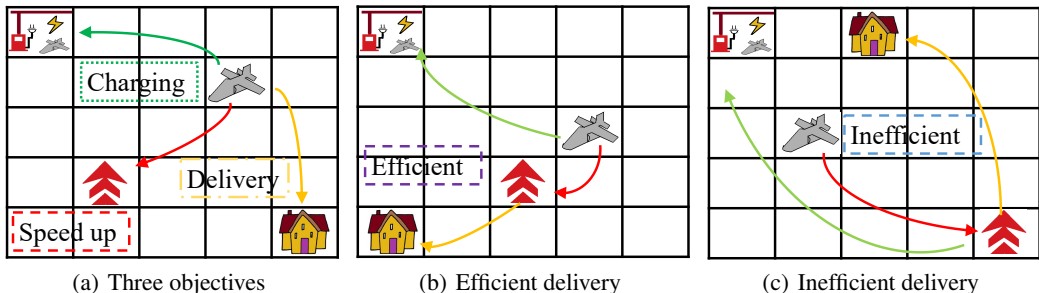

(a) Three objectives      (b) Efficient delivery      (c) Inefficient delivery

Figure 2: A sketch of Efficient Delivery. The purpose of the agent is to control the UAV to delivery as many packages as possible. When the current delivery location is reached, the next delivery location appears. In order to accomplish the delivery task more efficiently, it is necessary for the agent to weigh the importance of delivery, charging and acceleration at every moment.

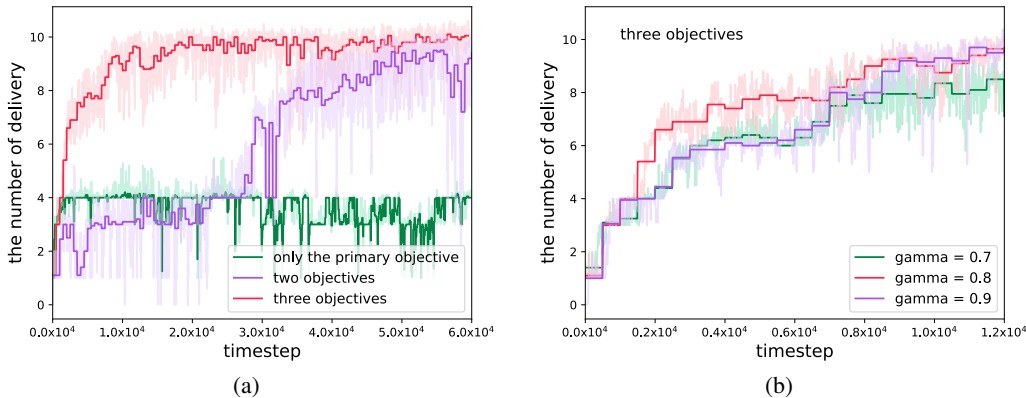

Figure 3: Performance comparison on Efficient Delivery games. (a) We compare our algorithm using all three objectives (delivery, charging and acceleration) with that using two objectives (delivery and charging), as well as DQN, which uses the reward function from delivery only. (b) We compare the performance of our algorithm under different discount factor $\gamma$.

importance of acceleration to the task, such as helping the agent charge and deliver faster, but also identifies that our algorithm can effectively learn the trade-off among multiple objectives. No matter using two or three objectives, our algorithm performs significantly better than DQN, which is simply optimized through a reward from delivery only. There are mainly two reasons. One reason is the delayed reward in DQN. Although the agent is charged, it does not receive any rewards. Therefore, it is difficult to find the relationship among these objectives, which is essential to understanding the task. The other reason is the myopic policy in DQN. Because of misalignment between the behavior and the cumulative discount return, the agent focuses mainly on high but short-term return. However, through multi-objectivization, the learned reward function is better shaped. The learning procedure of weight vector assigns preferences to additional objectives that are typically helpful for more delivery and corrects the relationships between the behavior and discount return.

**Algorithm Robustness Testing.** Some evidence (Prokhorov & Wunsch, 1997; Bertsekas & Tsitsiklis, 1996) show that the discount factor $\gamma$ has a great influence on the performance of RLs. Smaller $\gamma$ may lead to faster convergence, but poorer sub-optimal policy. However, in our framework, as Theorem 1 proves, different $\gamma$s induce the common optimal policy. Figure 3(b) shows the performance of our algorithm under different discount factors $\gamma$. The performances under three different settings are proximately the same, even though there is subtle fluctuation. The change of $\gamma$ by $\sim 20\%$ results in the change of the algorithm performance by $\sim 15\%$. Compared to the work of Xu et al. (2018), where the change of $\gamma$ by only $\sim 5\%$ results in a large change of performance by $\sim 30\%$, our algorithm is highly robust against $\gamma$. Therefore, the difficulty of myopic policy can be effectively alleviated in our framework.

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
