# OpenReview forum: "Solving single-objective tasks by preference multi-objective reinforcement learning"
_ICLR.cc/2020/Conference — Reject_

### Official Review · AnonReviewer2 · 2019-10-23
**Official Blind Review #2**

**Rating:** 1

**Review:**

After Responses:
I understand the differences that authors pointed to the relevant literature. However, it is still lacking comparisons to these relevant methods. The proposed method has not been compared with any of the existing literature. Hence, we do not have any idea how does it stand against the existing approaches. Hence, I believe the empirical study is still significantly lacking. I will stick to my decision. Main reason is as follows; I believe the idea is interesting but it needs a significant empirical work to be published. I recommend authors to improve empirical study and re-submit.
-------
The submission is proposing a method for multi-objective RL such that the preference of tasks learned on the fly with the policy learning. The main idea is converting the multi-objective problem into single objective by scalar weighting. The weights are learned in a structured learning fashion by enforcing them to approximate the Pareto dominance relations.

The submission is interesting; however, its novelty is not even clear since authors did not discuss majority of the existing related work.

Authors can consult the AAMAS 2018 tutorial "Multi-Objective Planning and Reinforcement Learning" by Whiteson&Roijers for relevant papers. It is also important to note that there are other methods which learn weighting. Optimistic linear support is one of such methods. Hence, this is not the first of such approaches. Beyond RL, it is also studied extensively in supervised learning. For example, authors can see "Multi-Task Learning as Multi-Objective Optimization" from NeurIPS 2018.

The manuscript is also very hard to parse and understand. For example, Definition 2 uses but not define "p" in condition (2). Similarly, Lemma 1 states sth is "far greater" than something else. However, "far greater" is not really defined. I am also puzzled to understand the relevance of Theorem 1. It is beyond the scope of the manuscript, and also not really new.

Authors suggest a method to solve multi-objective optimization. However, there is no correctness proof. We do not know would the algorithm result in Pareto optimal solution even asymptotically. Arbitrary weights do not result in Pareto optimality.

Proposing a new toy problem is well-received. However, not providing any experiment beyond the proposed problem is problematic. Authors motivate their method using DOOM example. Why not provide experimental results on a challenging problem like DOOM?

In summary, I definitely appreciate the idea. However, it needs better literature search. Authors should position their paper properly with respect to existing literature. The theory should be revised and extended with convergence to Pareto optimality. Finally, more extensive experiments on existing problems comparing with existing baselines is needed.

**Experience Assessment:**

I have published one or two papers in this area.

**Review Assessment: Checking Correctness Of Derivations And Theory:**

I carefully checked the derivations and theory.

**Review Assessment: Checking Correctness Of Experiments:**

I assessed the sensibility of the experiments.

**Review Assessment: Thoroughness In Paper Reading:**

I read the paper thoroughly.

---

> ### Author Response · Authors · 2019-11-13
> **Author response to Reviewer #2 (part 1)**
>
> We appreciate the time you spent reviewing our submission and hope our response help address some of your concerns.
>
> We believe the reviewer may have misunderstanded our paper to some extent. Different from the reviewer's summary "The main idea is converting the multi-objective problem into single objective by scalar weighting", we propose to transform a single-objective problem to a preference multi-objective problem with learnable dynamic weights. This main idea has been elaborated in the third paragraph of introduction.
>
> Q1: "its novelty is not even clear since authors did not discuss majority of the existing related work."
> A: Similar to MORL, the problem proposed by us also requires a learning agent to optimize two or more objectives at the same time. The major differences are as follows. 1) The problem setting is different. We focus on problems with one single primary objective and several additional helper-objectives, in which the main concern is how to utilize the helper-objectives so that the primary objective can be more efficiently optimized. 2) The weight vector is time-variant. Different from most MORL algorithms, where the weight vector is time-invariant, in our Efficient Delivery environment, in order to deliver as many packages as possible, it is necessary for the agent to weigh the importance of the three objectives in every state, according to the distances from the delivery location, the charging location and the acceleration location.
>
> In terms of the time-variant weight setting, there are two existing researches [1,2] that are somewhat similar to our work. However, the weights in these works are known in advance, and the research in this field mainly focuses on how to leverage transfer learning techniques to accelerate the learning process when the weights change over time. In contrast, the weights in our work are not clear and even ambiguous in the beginning of training, and the main idea is to let the agent learn the weights among these objectives. More vividly, the task in our setting can be considered as a series of multiple tasks with different weights in dynamic weights setting. Optimistic linear support[3] uses the concept of corner weights to pick the weights to use for creating scalarised instances. However, it is unknown to determine which corner weight to use by satisfying the users’ preferences. Moreover, each corner weight is still time-invariant. The work[4] explicitly cast multi-task learning as multi-objective optimization, with the overall objective of finding a Pareto optimal solution. A set of learnable weights is used to adjust the gradient magnitude from each task to weigh the conflict between tasks. Unlike this, the weight in our work is a function of state-action pair, thus it is time-variant.
>
> Q2: "Definition 2 uses but not define 'p' in condition (2)."
> A: We are sorry for the mistake. The letter 'p' is the initial of primary, denoting the primary objective.
>
> Q3: "Lemma 1 states sth is 'far greater' than something else. However, 'far greater' is not really defined."
> A: The 'far greater' is used in the expression '$r(s_0^1,a_0^1)+\cdots + r(s_{k-1}^1,a_{k-1}^1) \gg r(s_0^2,a_0^2)+\cdots + r(s_{t-1}^2,a_{t-1}^2)$', which is between Definition 1 and Definition 2. The symbol $'\gg'$ denotes our learning objective, taking '$a \gg b$' for example, which can be satisfied by maximizing $\left(a-b\right)$.

---

> > ### Author Response · Authors · 2019-11-13
> > **Author response to Reviewer #2 (part 2)**
> >
> >
> > Q4: "I am also puzzled to understand the relevance of Theorem 1."
> > A: In the process of weight optimization, the form of undiscounted cumulative return is adopted. However, the goal of the agent is to find a policy that maximizes the discounted cumulative return. Therefore, Theorem 1 proves that the optimal policy can also maximize the cumulative reward, although the reward fucntion is learned in the form of undiscounted accumulation. The recent work[5] also used undiscounting to optimize the reward fucntion, but it did not prove its rationality in theory, only showing its feasibility through many experiments, while we have proved its feasibility in theory for the first time.
> >
> > Q5: "there is no correctness proof."
> > A: The research of this paper can be seen as a kind of enrichment to the research on preference-based reinforcement learning[6], in which, as far as we have been concerned, no strict convergence proof has been provided. Here we provide a brief proof based on evolutionary algorithms. It is obvious that the optimal trajectory, which fits the preference Pareto dominance relations, can be sampled when an exploration rate $\sigma > 0$ is used during interaction with the environment. Meanwhile, the trajectory buffer always maintains the optimal trajectories found over time. Given the two points above, the optimal weight vector can be acquired through minimizing the cross-entropy loss (formula 7). For more details, please refer to the proof of convergence of evolutionary algorithms[7].
> >
> > Q6: "Why not provide experimental results on a challenging problem like DOOM?"
> > A: We agree that more experiments are helpful to show the effectiveness of our method. However, due to the limitations of time and computation resources, more experiments are left as future work.
> >
> > References:
> >
> > [1] Natarajan S, Tadepalli P. Dynamic preferences in multi-criteria reinforcement learning. In ICML, 2005.
> > [2] Abels A, Roijers D M, Lenaerts T, et al. Dynamic Weights in Multi-Objective Deep Reinforcement Learning. In ICML, 2019.
> > [3] Mossalam H, Assael Y M, Roijers D M, et al. Multi-objective deep reinforcement learning. arXiv preprint arXiv:1610.02707, 2016.
> > [4] Sener O, Koltun V. Multi-task learning as multi-objective optimization. In Advances in Neural Information Processing Systems, 2018.
> > [5] Christiano P F, Leike J, Brown T, et al. Deep reinforcement learning from human preferences. In Advances in Neural Information Processing Systems, 2017.
> > [6] Wirth C, Akrour R, Neumann G, et al. A survey of preference-based reinforcement learning methods. The Journal of Machine Learning Research, 2017, 18(1): 4945-4990.
> > [7] Rudolph G. Convergence properties of evolutionary algorithms. Kovac, 1997.

---

### Official Review · AnonReviewer3 · 2019-10-28
**Official Blind Review #3**

**Rating:** 3

**Review:**

Thank the authors for the response. I agree with R2 that the paper lacks comparisons with previous works. I will stick to my previous decision.
----------------------------------------
Summary
This paper presents a new approach for single-objective reinforcement learning by preferencing multi-objective reinforcement learning. The general idea is to first figure out a few important objectives, add some helper-objectives to the original problem, and learn the weights for each individual objective by trying to keep the same order as Pareto dominance. This paper has potential, but I lean to vote for rejecting this paper now, since it is still not ready. I might change my score based on the reviews from other reviewers.
Strengths
- The idea is novel. Learning weights for each objective by keeping the order as Pareto dominance is an interesting idea to me.
Weaknesses
- The lack of experiments. The authors tested their method in only one scenario, which makes me feel unsafe. Only testing on one simple scenario does not demonstrate the effectiveness. The authors are supposed to test their method on more (complex) scenarios to show the effectiveness of their method.
Possible Improvements
As mentioned before, the proposed method can be tested on more scenarios (e.g., Deep Sea Treasure, SuperMario, etc.).

**Experience Assessment:**

I have read many papers in this area.

**Review Assessment: Checking Correctness Of Derivations And Theory:**

I assessed the sensibility of the derivations and theory.

**Review Assessment: Checking Correctness Of Experiments:**

I assessed the sensibility of the experiments.

**Review Assessment: Thoroughness In Paper Reading:**

I read the paper at least twice and used my best judgement in assessing the paper.

---

> ### Author Response · Authors · 2019-11-13
> **Author response to Review #3**
>
> Thank you for providing the feedback. We hope the following address some of your concerns.
>
> We agree that more experiments are helpful to show the effectiveness of our method. However, almost all benchmark scenarios (e.g., Deep Sea Treasure, SuperMario, etc), which have been widely used to measure the performance of MORL algorithms, are not suitable for evaluating our contribution. The reasons are as follows. 1) The problem setting is different. Although the problem proposed also requires a learning agent to optimize two or more objectives at the same time, the major difference is that we focus on problems with one single primary objective and several additional helper-objectives, in which the main concern is how to utilize the helper-objectives so that the primary objective can be more efficiently optimized. 2) The weight vector is time-variant. Different from most benchmark scenarios, where the weight vector is time-invariant, in our Efficient Delivery environment, in order to deliver as many packages as possible, it is necessary for the agent to weigh the importance of the three objectives in every state, according to the distances from the delivery location, the charging location and the acceleration location.
>
> In addition, the feasibility of our algorithm has been theoretically analyzed. Although only one scenario was used, we believe that the effectiveness of our algorithm has been sufficiently demonstrated. Therefore, we sincerely hope that the reviewer can understand why only one scenario was used in this paper.

---

### Decision · Program_Chairs · 2019-12-19

**Decision:**

Reject

**Comment:**

The paper considers planning through the lenses both of a single and multiple objectives. The paper then discusses the pareto frontiers of this optimization. While this is an interesting direction, the reviewers feel a more careful comparison to related work is needed.